# Report on the Effect of the Implementation of an Early Detection and Prevention of Cancer Program on Families at High Hereditary Risk—Concentrating on Patients Undergoing Genetic Diagnostics and Counseling in Central Poland

**DOI:** 10.3390/ijms241713178

**Published:** 2023-08-24

**Authors:** Tadeusz Kałużewski, Izabela Kubiak, Michał Bednarek, Jordan Sałamunia, Dorota Kucharska, Łukasz Kępczyński, Marek Stempień, Tobiasz Kubicki, Radzisław Trzciński, Zofia Gordon-Sönmez, Anna Bartosińska-Dyc, Agnieszka Gach, Bogdan Kałużewski

**Affiliations:** 1Department of Genetics, Polish Mother’s Memorial Hospital Research Institute, 93-338 Lodz, Poland; lukasz.kepczynski@iczmp.edu.pl (Ł.K.);; 2R&D Department, Laboratory of Medical Genetics, GENOS Sp. z o.o., 91-033 Lodz, Polandb.kaluzewski42@gmail.com (B.K.); 3Regional Center for Cancer Prevention, Zduńska Wola District Hospital Sp. z o.o., 98-220 Zdunska Wola, Poland; 4Department of General and Colorectal Surgery with the Subdepartment of Oncological Surgery, Provincial Hospital John Paul II in Bełchatów, 97-400 Belchatow, Polandtrzcinskir@wp.pl (R.T.); 5Sokrates Medical Center, 98-200 Sieradz, Poland; 6Institute of Medical Sciences, Collegium Medium of the Jan Kochanowski University in Kielce, 25-369 Kielce, Poland; 7Department of Gynecology and Obstetrics, Zduńska Wola District Hospital Sp. z o.o., 98-220 Zdunska Wola, Poland

**Keywords:** hereditary breast ovarian cancer, breast cancer, ovarian cancer, hereditary colorectal cancer, colorectal cancer, hereditary endometrial cancer, endometrial cancer, *BRCA1*, *BRCA2*, *CHEK2*, *PALB2*

## Abstract

Over a 46-month period, the objectives of the National Cancer Control Program (NCCP, pol. Narodowy Program Zwalczania Chorób Nowotworowych), coordinated by the Ministry of Health, were pursued by conducting genetic diagnostics on individuals at high risk of developing cancer. A total of 1097 individuals were enrolled in the study, leading to the identification of 128 cases of germline mutations. The implementation of the NCCP led to the identification of genetic mutations in 4.43% of the patients qualified for *BRCA1* and *BRCA2* screening tests, in 18.18% of those qualified for a comprehensive next-generation sequencing (NGS) panel in cases of breast and ovarian cancer, and in 17.36% of cases of colorectal and endometrial cancer. The research conducted allowed us to establish individualized preventive and therapeutic approaches for mutation carriers. However, the results prove that liberalizing the inclusion criteria for high-throughput diagnostics and the use of broad gene panels could significantly increase the percentage of detected carriers. This publication serves as a summary and discussion of the results obtained from the implementation of the NCCP as well as of the role of genetic consulting in personalized medicine.

## 1. Introduction

Cancer is a diverse group of diseases, constituting one of the most significant public health concerns worldwide. While most cancer cases are sporadic, a hereditary form linked to genetic mutations occurs in a significant number of patients. The most common cases of hereditary cancer include breast, ovarian, colon, and uterus cancers. Understanding the genetic basis of these cancers is crucial for risk assessment, early detection, and targeted treatment for individuals prone to these diseases. Genetic testing and counseling play a significant role in identifying individuals at increased risk and implementing appropriate surveillance programs. The starting point for the formulation of this study’s objectives was the identification of founder mutations in the *BRCA1* and *BRCA2* genes. These mutations are characteristic of the population of females affected by breast and/or ovarian cancers in the north-western region of Poland [1]. The remarkable popularization of *BRCA1* mutation diagnostics through a relatively simple and cost-effective polymerase chain reaction (PCR) test is also worth emphasizing [2]. After over two decades of self-promotion, the topic of preventive research gained recognition and found funding sources within the framework of the National Cancer Control Program (NCCP) coordinated by the Ministry of Health; this framework consisted of two modules: “Module I—early detection of malignant tumors in families at high, hereditary risk of breast and ovarian cancer” and “Module II—early detection and prevention of malignant tumors in families at high, hereditary risk of colorectal and endometrial cancer”. The Program’s scope was developed by a group of experts, appointed by the Ministry of Health, and included detailed guidelines on patient qualifications, diagnostic methods, and preventive and therapeutic care. The aims of our study were to disseminate the research results and discuss the qualification criteria and the accepted methodology. In addition, we have presented the results of tests performed optionally, i.e., outside the program guidelines, and their effect on the percentage of detected mutation carriers.

## 2. Results

### 2.1. Module I

A total of 953 patients were enrolled in Module I, 141 of whom were classified as the highest-risk group and 372 as the high-risk group. An overview of the results is presented in Table 1. Screening tests for *BRCA1* and *BRCA2* mutations were performed on 880 individuals, leading to the identification of 38 *BRCA1* mutations and 1 *BRCA2* mutation. In total, 843 patients were included in the *CHEK2* and *PALB2* gene mutation tests, wherein nine *CHEK2* and three *PALB2* mutations were revealed. Following the execution of targeted familial diagnostics, 19 carriers of *BRCA1* mutations, 7 carriers of *BRCA2* mutations, 2 carriers of *CHEK2* mutations, and 3 carriers of *PALB2* mutations were identified. Complete sequencing of the *BRCA1* and *BRCA2* genes was conducted for 55 patients; consequently, five *BRCA1* and two *BRCA2* mutations were detected. Additionally, in the assessment of the point mutations beyond the scope of the NCCP, one *RAD51C* mutation, one *CDKN2A* mutation (a variant of conflicting interpretations; thus, it is not listed in Table 1), two *MUTYH* mutations (heterozygous), one *CHEK2* mutation (a variant of uncertain significance), and two *PALB2* mutations were detected, all of which were not covered by the screening examination. The details of the pathogenic and likely pathogenic mutations detected by the next-generation sequencing (NGS) technique are presented in Table 2. In cases where the pedigree-clinical criteria were met and the patients did not qualify for NGS testing (or the test yielded negative results), the diagnostic process was expanded to include multiplex ligation-dependent probe amplification (MLPA) analysis, resulting in the identification of five mutations in 76 eligible patients.

### 2.2. Module II

A total of 144 patients were enrolled for genetic testing in Module II. An overview of the detected pathogenic mutations is presented in Table 3. Molecular diagnoses were established in 25 cases (17.36%). We identified eleven mutations in the *APC* gene, which is responsible for familial adenomatous polyposis; five mutations in the *MLH1* and *MSH6* genes, which are associated with Lynch syndrome; one mutation in the *STK11* gene, responsible for the Peutz–Jeghers syndrome; and one heterozygous mutation in the *MUTYH* gene (which is responsible for the recessive polyposis syndrome, thus not listed in Table 3). By employing appropriate evaluative procedures within Module II, the study’s diagnostic scope was expanded to include mutations not specified in the detailed program description. Additionally, two *BRCA1* mutations, one *BRCA2* mutation, one *PALB2* mutation, one *NOD2* mutation (the variant with conflicting interpretations), one *NBN* mutation, and three *ATM* gene mutations were identified.

## 3. Discussion

Breast cancer is the most common malignant tumor among women in Poland, accounting for 22.9% of all cancer cases. It also ranks the second leading cause of cancer-related deaths, after lung cancer, accounting for 15.1% of deaths [3]. It is estimated that approximately 5-10% of breast cancers have a hereditary basis that can be identified using the current diagnostic methodology [4]. Ovarian cancer is responsible for 4.3% of female cancers and ranks fifth in terms of incidence and fourth in terms of mortality rate (6.0%) [3]. Germ-line mutations can be identified in approximately 25% of ovarian cancers [4]. The most common cause of hereditary breast and ovarian cancers (HBOC) is the presence of mutations in the *BRCA1* and *BRCA2* genes. In addition, there are high-risk genes, (which are associated with other predisposition syndromes), such as *CDH1*, *PALB2*, *PTEN*, *STK11*, and *TP53*, as well as moderate-risk genes, including *ATM*, *BARD1*, *BRIP1*, *CHEK2*, *EPCAM*, *MLH1*, *MSH2*, *MSH6*, *PMS2*, *RAD51C*, and *RAD51D*. In addition to these genes, a high risk for ovarian cancer is associated with mutations in the *DICER1*, *VHL*, *PTCH1*, *SUFU*, *SMARCB1*, and *SMARCA4* genes. Colorectal cancer is the third most commonly diagnosed malignant tumor among men (accounting for 6.8% of all cases) and the fourth most common among women (5.9%). It is the third leading cause of death for both men and women (8.0% and 7.7%, respectively). Approximately 5% of colorectal cancer patients develop the disease due to a hereditary predisposition, with the most common types being Lynch syndrome (approximately 2–4% of patients) and familial adenomatous polyposis (approximately 1% of patients) [4]. Endometrial cancer is the third most common cancer in women (7.0%) and the sixth leading cause of death (4.0%). Similar to colorectal cancer, Lynch syndrome is the most common inducer of familial risk for endometrial cancer [4].

The tasks of the National Cancer Control Program, coordinated by the Ministry of Health, aim to provide specialized prevention measures for families at increased familial risk. The introduction of the program in 2018 was a breakthrough for genetic diagnostics in this area in Poland, and it also increased our knowledge about the role of genetic factors in the development of genetic diseases. The educational aspect of this program, i.e., increasing the awareness of genetic testing and preventive measures among patients, is undoubtedly its key advantage. The results presented in this study indicate that a fairly high percentage of pathogenic mutation carriers was detected. Screening tests for founder *BRCA1*/*BRCA2* mutations yielded positive results for 4.43% of the qualified patients (15.60% in the highest-risk group); most of these mutations were founder mutations of the *BRCA1* gene, constituting a finding similar to results published earlier for the north-western region of Poland [1,5,6]. The simple qualification criteria employed in this study provide an additional benefit of the screening tests, enabling other specialists, like oncologists or gynecologists, to take their first steps into the domain of genetic tests. This may also be a crucial time-reduction factor for patients on waiting lists, which are usually rather long in the countries like Poland, where access to clinical geneticists is still rather limited.

It should, however, be noted that the diagnostic apparatus of the program did not identify all families with an increased cancer risk, although this should be possible considering today’s state-of-the-art technological potential. When describing the program’s results, it is important to mention that due to the additional involvement of our genetic facility, we were able to identify 16 families with high- or intermediate-risk mutations and who thus failed to meet the criteria for the tests in question. This result accounted for as much as 17.58% of all the detected mutations. During the implementation of the program, we identified two *PALB2* mutations and one *RAD51C* mutation as well as five deletions in *BRCA1*/*BRCA2* genes via MLPA, amounting to 12.12% of the positive results in Module I that could have remained undetected with the testing limitations assumed in the NCCP. In addition, two patients with *MUTYH* mutations were identified in Module I, thereby facilitating comprehensive genetic consulting for both the patients and their families. Also, in Module II, we identified two mutations in *BRCA1*, one mutation in *BRCA2*, one mutation in *PALB2*, three mutations in *ATM*, and one mutation in *NBN* in the patients who qualified for molecular testing for Lynch syndrome (and failed to meet either the criteria of the HBOC or the inclusion criteria for Module I). This amounts to 32% of the positive results in Module II. Considering the fact that the expansion of the diagnostic scope, specified in the program, yielded diagnostic results in a minority of cases, it may be assumed that the real number of undetected carriers of pathogenic mutations may be significantly higher. The PCR-based screening tests focused on specific DNA regions. At the same time, the NGS enabled the simultaneous sequencing of longer DNA fragments, which enabled the sequencing of multiple genes at once, thereby substantially increasing the throughput and scalability figures.

Therefore, offering a high-throughput diagnostic apparatus to a broader group of patients seems highly justified. The design of the program discussed herein focused on screening tests for economic reasons, as the cost of screening tests is several times lower than that of high-throughput tests. However, it is worth noting that the costs of comprehensive genetic diagnostics account for no more than 1% of the costs of the oncological care of affected patients. Hopefully, future cost evaluations will consider not only single cost areas but also long-term clinical benefits, such as higher percentages of effectively cured patients, more effective systemic treatments, shorter sick leave periods, and reduced cancer mortality rates. The hitherto-followed practice of adhering to the international recommendations of scientific societies should be expedited and further supplemented by a comprehensive cost analysis, encompassing all the elements of personalized prevention, diagnostics, potential therapy, and rehabilitation leading to health maintenance. A case in point could involve the augmentation (or omission) of specific molecular tests in favor of high-coverage next-generation sequencing procedures. Another pressing concern pertains to horizontal integration, which more profoundly accounts for collaboration among the diverse medical realms (with various medical specialties) engaged in the prevention, diagnostics, and treatment of neoplastic ailments.

On the other hand, high-throughput testing creates problems regarding interpretation. During the implementation of the program discussed herein, we came across conflicting interpretations (*CDKN2A* NM_000077.5:c.442G>A, *NOD2* NM_022162.2:c.3019dup, and *CHEK2* NM_007194.4 c.470T>C), a variant of uncertain significance (*CHEK2* NM_007194.4 c.1211A>G), and a variant never reported (the NM_000059:4:c.7990_7991del in *BRCA2* gene). Despite advances in sequencing technologies, a method for distinguishing between such variants’ pathogenic and benign characteristics remains elusive. Integrating diverse data sources and population databases is crucial for accurate classification, enhancing precision in clinical decision making. It is necessary to confer with highly qualified genetic consultants to explain these variants to patients. Additionally, identifying such variants emphasizes the significance of functional studies, which are rarely performed in cases of hereditary syndromes but are crucial to clarify the meaning of the detected variants.

## 4. Materials and Methods

### 4.1. Patient Qualification—Module I

The patients who attended our clinic were most often residents of the Voivodeship of Lodz, where our facility is located. Due to the high diagnostic potential of our clinic, we also received patients from neighboring voivodeships, especially from Mazovia, Greater Poland, Silesia, Lower Silesia, and Lesser Poland. The criteria proposed in the NCCP were used in the process of qualifying patients for Module I. A group of individuals at high and highest risks were identified based on the division outlined in the detailed description of the program employed. In ambiguous situations, the criteria for hereditary breast-ovarian cancer syndrome (HBOC) were followed, as outlined in the National Comprehensive Cancer Network (NCCN) guidelines (https://onlinelibrary.wiley.com/doi/10.1002/cam4.2534, accessed on 21 August 2023). Regardless of a patient’s program eligibility status, the specific criteria for inclusion in genetic testing were applied.

#### 4.1.1. Qualification Criteria for the Highest-Risk Group

The individuals who qualified for inclusion in the highest-risk group were as follows:Individuals from families with 3 or more cases of breast cancer and/or ovarian cancer among first- and second-degree relatives (including the proband);Individuals identified with a pathogenic mutation in the *BRCA1*, *BRCA2*, or *PALB2* genes, regardless of their family history.

#### 4.1.2. Qualification Criteria for the High-Risk Group

A high risk of developing breast and/or ovarian cancer (at least 4-5 times higher than that in the general population) qualified a given patient for inclusion in the high-risk group, for which the following clinical criteria were applied:Families with 2 cases of breast and/or ovarian cancer among the proband or first- and second-degree relatives (or 2 cases among second- and third-degree relatives on the paternal side), particularly when at least one affected individual had been diagnosed with ovarian cancer and one case of said cancer had occurred before the age of 50.Families with bilateral breast cancer diagnosed in first- and second-degree relatives.Families with breast cancer diagnosed before the age of 40 in first- and second-degree relatives.Families with breast cancer diagnosed in males among first- and second-degree relatives.

#### 4.1.3. Qualification for Genetic Testing

The primary screening tests for the 5 most common mutations in the *BRCA1* gene in the Polish population included the following individuals:All individuals diagnosed with ovarian/fallopian tube/peritoneal cancer;All individuals diagnosed with breast cancer;First- and second-degree relatives of individuals with breast and/or ovarian cancer for whom marker mutations could not be established and diagnostics could not be employed for the affected individual.

The basic screening tests for the 3 most common protein-truncating mutations in the *CHEK2* gene and 2 mutations in the *PALB2* gene in the Polish population were designed to include the following individuals:All individuals diagnosed with breast cancer;First-degree relatives of individuals with breast cancer from families meeting the criteria for high and the highest risk of breast cancer.

Only individuals who had been diagnosed with breast and/or ovarian cancer without presenting any of the 5 most common *BRCA1* mutations were qualified to participate in the examination of *BRCA1* and *BRCA2* mutation carrier status using next generation sequencing (NGS), provided that the following conditions were met:The affected individual had been diagnosed with breast cancer or ovarian cancer and had at least 2 first- or second-degree relatives with a diagnosis of breast and/or ovarian cancer, where at least one of such cases had occurred before the age of 50;The affected individual had been diagnosed with breast cancer before the age of 50 or ovarian cancer at any age and had a first- or second-degree relative who had been diagnosed with breast cancer (breast cancer in males) and/or ovarian cancer;The same affected individual had been diagnosed with both breast and ovarian cancer or bilateral breast cancer, including at least one case below the age of 50;The affected individual had been diagnosed with ovarian cancer and had at least one relative with breast cancer—which had been diagnosed before the age of 50—or who had been diagnosed with ovarian cancer.

Furthermore, all the family members for whom the highest or high-risk mutation was identified were eligible for examinations to determine their carrier status.

### 4.2. Patient Qualification—Module II

A clinical geneticist enrolled the patients based on their detailed family histories (including information on all their relatives, such as the age at the onset of cancer and the type/location of tumors among those relatives, as well as data on unaffected relatives). No clinical criteria were defined in the program for hereditary gastrointestinal cancer syndromes; therefore, the patients were qualified individually by the consulting clinical geneticists.

### 4.3. Laboratory Methodology

By exploiting the funding available for the screening tests, the following five mutations, most frequently occurring in the *BRCA1* gene within the Polish population, were identified: c.68_69del, c.4035del, c.5266dup, c.3700_3704del, and c.181T>G [7]; these mutations accounted for approximately 62% of all the identified mutations in the *BRCA1* gene in Poland [8]. Additionally, the following three most common protein-truncating mutations in the *CHEK2* gene were identified: 1100del, IVS+1G>A, and del5395. Furthermore, the two most common mutations in the *PALB2* gene were identified, namely, c.509_510del and c.172_175del [9]. An overview of the mutations identified in screening tests are presented in Table 4 and Table 5.

Screening for mutations in the *BRCA1* gene was performed using the R-27/P-48FRT Oncogenetics BRCA Panel based on a Real-Time PCR reaction (Sacace Biotechnologies, Como, Italy). A Real-Time PCR (RT-PCR) reaction was carried out using the Qiagen RotorGene Q instrument (Qiagen GmbH, Hilden, Germany). An analysis of the obtained variants was performed by means of the Rotor-Gene Q 2.1.0.9 software (Qiagen GmbH, Hilden, Germany). The use of the aforementioned method enabled the detection of the five most common mutations included in Module I of the program as well as two additional mutations in the *BRCA1* gene and one mutation in the *BRCA2* gene.

Conventional Sanger sequencing and allele-specific amplification PCR (in case of the del5395 mutation) were employed to screen the most common mutations in the *CHEK2* and *PALB2* genes.

Moreover, conventional Sanger sequencing was the method of choice for familial variant carrier testing, when the familial variant carriers were not detectable, using the rapid RT-PCR test. The complete sequencing of the *BRCA1* and *BRCA2* genes using next-generation sequencing was outsourced to subcontractors. Whenever possible and indicated to be required, the diagnostic process was supplemented outside of the NCCP by examining extensive deletions and duplications in the *BRCA1* and *BRCA2* genes, for which the MLPA method with SALSA MLPA Probemix P002-D1-0918 and SALSA MLPA Probemix P045 D1-0519 kits was used, and the complete sequencing of other genes, correlated with cancer predisposition, besides the *BRCA1* and *BRCA2* genes was employed (these genes included *AKT1*, *APC*, *ATM*, *AXIN2*, *BARD1*, *BMPR1A*, *BRCA1*, *BRCA2*, *BRIP1*, *CDC73*, *CDH1*, *CDKN1B*, *CDKN2A*, *CHEK2*, *CTNNA1*, *DICER1*, *EPCAM*, *FANCC*, *FH*, *GALNT12*, *GDNF*, *GREM1*, *HNF1A*, *HNF1B*, *HOXB13*, *KIF1B*, *MAX*, *MC1R*, *MEN1*, *MET*, *MITF*, *MLH1*, *MLH3*, *MRE11*, *MSH2*, *MSH6*, *MUTYH*, *NBN*, *NF1*, *PALB2*, *PIK3CA*, *PMS2*, *POLD1*, *POLE*, *POT1*, *PRKAR1A*, *PRSS1*, *PTCH1*, *PTEN*, *RAD51C*, *RAD51D*, *RB1*, *RET*, *SDHA*, *SDHAF2*, *SDHB*, *SDHC*, *SDHD*, *SMAD4*, *STK11*, *TERT*, *TGFBR2*, *TMEM127*, *TP53*, *TSC1*, *TSC2*, *VHL*, *WT1*, *XRCC2*, and *XRCC3*).

In Module II, the same panel of 70 genes (including the detailed analysis of genes such as *APC*, *MLH1*, *MSH2*, *MSH6*, *PMS2*, *STK11*, *SMAD4*, *BMPR1A*, *EPCAM*, and *MUTYH*) was used for diagnostic purposes, using next-generation sequencing technology, together with MLPA kits, to identify deletions and duplications in the genes associated with specific clinical diagnoses.

The scheme behind the inclusion of molecular methods in the diagnostic process has been depicted using a block chart (Figure 1). It is important to note that not every patient underwent the whole process; some had selected tests performed in the past or could not participate in consecutive steps.

### 4.4. Editorial Policy and Ethical Considerations

The authors obtained written informed consent for clinical genetic testing and anonymous publication of results from the patients in accordance with applicable local laws. In the case of patients tested in NCCP, consent forms were provided by the Ministry of Health; in other cases, an interior declaration was used. This study was approved by the Bioethics Committee of the Polish Mother’s Memorial Hospital Research Institute (approval number 80/2017). All the procedures performed in this study followed the principles of the Declaration of Helsinki.

## 5. Conclusions

In conclusion, the prevalence of hereditary mutations in breast, ovarian, colorectal, and endometrial cancers highlights the significance of molecular testing and genetic consulting with respect to these malignancies. The National Cancer Control Program has provided specialized preventive measures for families at elevated familial risk and increased patient awareness regarding genetic testing and preventative strategies. This study was performed using a representative group of 1097 individuals, constituting a significant quantity with which to draw meaningful conclusions. The program’s results demonstrate that a considerable percentage of pathogenic mutation carriers was identified, underscoring the benefits of screening tests, particularly for founder *BRCA1* mutations. This program’s simplified qualification criteria offer an additional advantage by enabling specialists from various fields to initiate genetic testing, which has expanded the availability of diagnostics. On the other hand, the referral of patients by different specialists hindered the consistency of the diagnostics workflow and caused the placement of genetic consultations in different stages of the therapeutic process. The expansions of the diagnostic scope revealed many other mutations that could have been missed under the employed program’s criteria. The additional involvement of our facility allowed us to demonstrate the inadequacies of the NCCP’s assumptions. In this context, the emergence of high-coverage next-generation sequencing holds great potential. While economic considerations initially favored screening tests, the long-term clinical benefits of comprehensive genetic diagnostics, including increased patient cure rates, more effective treatments, and reduced cancer mortality rates, warrant comprehensive cost evaluations. Nevertheless, challenges in interpreting high-throughput data highlight the need for highly qualified genetic consulting to navigate variant interpretations and uncertainties.

## Figures and Tables

**Figure 1 ijms-24-13178-f001:**
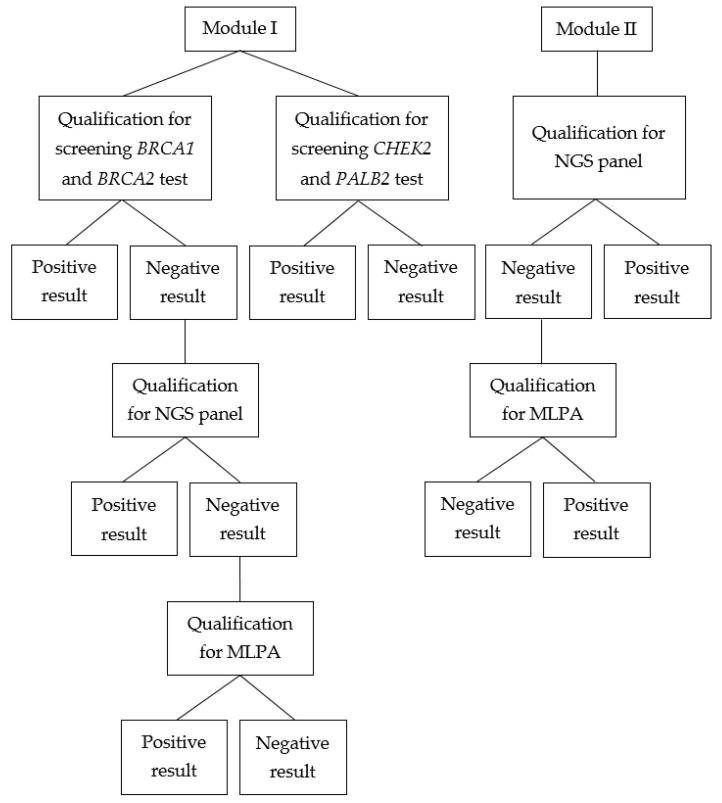
Block chart presenting qualification for genetic testing and diagnostic process.

**Table 1 ijms-24-13178-t001:** Overview of selected tests performed within Module I.

Test	Number of Tests	Number of Pathogenic Mutations	Percentage of Pathogenic Mutations
*BRCA1* and *BRCA2* screening test	880	39	4.43%
*Highest-risk group*	141	22	15.60%
*High-risk group*	372	11	2.96%
*CHEK2* screening test	843	9	1.07%
*Highest-risk group*	84	9	10.71%
*High-risk group*	329	0	0%
*PALB2* screening test	843	3	0.35%
*Highest-risk group*	84	0	0%
*High-risk group*	329	3	0.91%
MLPA *BRCA1*	70	3	4.29%
MLPA *BRCA2*	62	2	3.33%
Next-generation sequencing	55	10	18.18%

**Table 2 ijms-24-13178-t002:** Mutations detected using next-generation sequencing.

Gene	Mutation	dbSNP Number	ClinVar Submissions	Internal Classification
*BRCA1*	NM_007294.4: c.4986+6T>G	rs80358086	12 (P), 2 (LP)	Pathogenic
*BRCA1*	NM_007294.4:c.81-2A>C	rs397509326	2 (P), 2 (LP)	Pathogenic
*BRCA1*	NM_007294.4:c.115T>C	rs80357164	15 (P), 1 (VUS)	Pathogenic
*BRCA1*	NM_007294.4:c.5034_5037del	rs80357580	27 (P)	Pathogenic
*BRCA1*	NM_007294.4:c.2761C>T	rs80357377	10 (P)	Pathogenic
*BRCA2*	NM_000059.4:c.3975_3978dup	rs397515636	23 (P)	Pathogenic
*BRCA2*	NM_000059:4:c.7990_7991del	-	0	Likely pathogenic
*PALB2*	NM_024675.3:c.2962C>T	rs118203999	8 (P)	Pathogenic
*PALB2*	NM_024675.3:c.697del	rs180177090	6 (P)	Pathogenic
*RAD51C*	NM_058216.3:c.577C>T	rs200293302	13 (P), 1 (LP)	Pathogenic

**Table 3 ijms-24-13178-t003:** Overview of mutations detected within Module II.

Gene	Mutation	dbSNP Number	ClinVar Submissions	Internal Classification	Number of Patients
*APC*	NM_000038.4:c.3927_3931del	rs121913224	30 (P), 1 (LP)	Pathogenic	5
*APC*	NM_000038.6:c.2626C>T	rs121913333	11 (P), 1 (LP)	Pathogenic	5
*APC*	NM_000038.6:c.4438C>T	-	1 (P)	Pathogenic	1
*MLH1*	NM_000249.4:c.83C>T	rs63750792	7 (P), 1 (LP)	Pathogenic	1
*MLH1*	NM_000249.4:c.1897-2A>G	rs267607871	4 (LP), 5 (VUS)	Pathogenic	1
*MSH6*	NM_000179:c.423del	rs1114167728	3 (P), 1 (LP)	Pathogenic	3
*STK11*	NM_000455.5:c.891G>T	rs730881984	1 (P)	Pathogenic	1
*BRCA1*	NM_007294.4:c.181T>G	rs28897672	56 (P)	Pathogenic	1
*BRCA1*	NM_007294.4:c.5251C>T	rs80357123	36 (P)	Pathogenic	1
*BRCA2*	NM_000059.4:c.3076A>T	rs80358552	8 (P)	Pathogenic	1
*PALB2*	NM_024675.4:c.759del	rs1060499830	2 (P), 1 (LP)	Pathogenic	1
*ATM*	NM_000051.4:c.742C>T	rs730881336	10 (P), 1 (LP)	Pathogenic	3
*NBN*	NM_002485.5:c.657_661del	rs587776650	36 (P), 1 (LP)	Pathogenic	1

**Table 4 ijms-24-13178-t004:** Overview of the mutations identified in the *BRCA1* and *BRCA2* genes during the mutation-screening test.

Gene	HGVS Variant	The Common Name of Mutation	Comment
*BRCA1*	NM_007294.3:c.5266dup	5382insC	Mutation included in the NCCP
*BRCA1*	NM_007294.3:c.181T>G	C61G	Mutation included in the NCCP
*BRCA1*	NM_007294.3:c.4035del	4153delA	Mutation included in the NCCP
*BRCA1*	NM_007294.3:c.68_69del	185delAG	Mutation included in the NCCP
*BRCA1*	NM_007294.3:c.3700_3704del	3819del 5	Mutation included in the NCCP
*BRCA1*	NM_007294.3:c.3756_3759del	-	Mutation investigated additionally.
*BRCA1*	NM_007294.3:c.1961del	-	Mutation investigated additionally.
*BRCA2*	NM_000059.3:c.5946del	-	Mutation investigated additionally.

**Table 5 ijms-24-13178-t005:** Overview of mutations identified in the *CHEK2* and *PALB2* genes during the mutation-screening test.

Gene	HGVS Variant	The Common Name of Mutation	Comment
*CHEK2*	NM_007194.3:c.444+1G>A	IVS+1G>A	-
*CHEK2*	NM_007194.3:c.1100del	1100delC	-
*CHEK2*	NM_007194.3:c.909-?_1095+?del	del 5395	-
*PALB2*	NM_024675.3:c.509_510del	-	-
*PALB2*	NM_024675.3:c.172_175del	-	-

## Data Availability

Due to the nature of this research and the corresponding ethical and legal considerations, supporting data are not available.

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
