# Peer review of "Report on the Effect of the Implementation of an Early Detection and Prevention of Cancer Program on Families at High Hereditary Risk—Concentrating on Patients Undergoing Genetic Diagnostics and Counseling in Central Poland"

_ijms, 2023, doi:10.3390/ijms241713178_

Round 1
Reviewer 1 Report
Tadeusz Kałużewski et al. aimed to establish personalized preventive and therapeutic strategies for individuals carrying mutations. They utilized PCR and next-generation sequencing techniques to conduct genetic diagnostics on individuals at high risk of developing cancers, particularly breast cancer. However, it is important to note that this study suffers from methodological confusion as different approaches were employed for genetic testing. Some patients underwent screening tests for BRCA1 and BRCA2 mutations, while others had complete sequencing of the BRCA1 and BRCA2 genes, and some underwent NGS testing. Unfortunately, the author did not provide an explanation for this variation in testing methods or clarify the differences in the results detected by these three approaches. Additionally, the author failed to provide sufficient background information and did not adequately reflect on the significance, value, and guiding implications of the research findings.

Numerous grammar issues have been identified. Please proceed with the necessary corrections and revisions to enhance clarity and readability.
Grammar issues:
Line 22-23: "preventive and therapeutic approaches to mutation carriers" should be "preventive and therapeutic approaches for mutation carriers."
Line 24: has a repetitive "of."
Line 32: "characteristic for" should be "characteristic of."
Line 33: "affected by breast and/or ovarian cancer" should be "affected by breast and/or ovarian cancers."
Line 45: "Out of whom" should be "Out of which."
Line 49: "In targeted familial diagnostics" should be "During targeted familial diagnostics."
Line 64: "As many as" is not incorrect, but it could be more concise and simply stated as "A total of."
Line 69-70: The phrase "Having employed appropriate evaluating procedures within Module II" should be "By employing appropriate evaluation procedures within Module II."
Line 88-89: "Apart from these genes, the high-risk for ovarian cancer" should be "In addition to these genes, high-risk genes for ovarian cancer."
Line 104-105: "increasing the awareness of genetic testing and preventive measures among patients, what is an invaluable advantage" should be "increasing the awareness of genetic testing and preventive measures among patients, which is an invaluable advantage."
Line 115: "which should be fairly possible" could be rephrased as "which should be possible."
Line 118: "When describing the Program results, it is important to mention that, thanks to the additional involvement of our genetic facility,," there is an unnecessary comma before the double comma at the end.
Author Response
Dear Reviewer,
We would like to thank you for your valuable comments. Below please find our comments to the issues raised in the review. All the changes, made in manuscript, have been marked in red. We are open to further improvement discussions, were the provided revisions not sufficient for any reason.
Review: Tadeusz Kałużewski et al. aimed to establish personalized preventive and therapeutic strategies for individuals carrying mutations.
Author comment: The presented study did not aim the above target. The personalized preventive and therapeutic strategy was established in the National Cancer Control Program, created by the Polish Ministry of Health. Our study aimed to present the results of the Program implementation at our Clinic and discuss the methodology given by the Program’s authors. In order to make the study goals clearer, we have changed the Introduction section from: “The aim of this report was to disseminate the research results, discuss the qualification criteria and the accepted methodology” to "The Program's scope was developed by a group of experts, appointed by the Polish Ministry of Health, and included detailed guidelines on patient qualifications, diagnostic methods, and preventive and therapeutic care. The aims of our study were to disseminate the research results, discuss the qualification criteria and the accepted methodology. In addition, we have presented the results of optional (off-Program) tests and their effect on the percentage of detected mutation carriers."
Review: However, it is important to note that this study suffers from methodological confusion as different approaches were employed for genetic testing. Some patients underwent screening tests for BRCA1 and BRCA2 mutations, while others had complete sequencing of the BRCA1 and BRCA2 genes, and some underwent NGS testing. Unfortunately, the author did not provide an explanation for this variation in testing methods or clarify the differences in the results detected by these three approaches.
Author comment: We have made every effort to ensure that the applied methodology, adopted in the NCCP, is transparent. We aimed the study to be understandable also for medical professionals not experienced in clinical genetics, therefore, we have indicated fragments of the materials and methods section that explain the issues scored by the Reviewer. Additionally, we have added Figure 1 to the Material and methods section, illustrating the diagnostic process by means of a block diagram to make it clearer. If the implemented fragments are not sufficient for the Reviewer we would highly appreciate if the Reviewer could specify what aspects should further be changed or added.
Review: Some patients underwent screening tests for BRCA1 and BRCA2 mutations, while others had complete sequencing of the BRCA1 and BRCA2 genes, and some underwent NGS testing.
Patients indeed underwent screening tests for BRCA1 and BRCA2 mutations, while some of them, who had qualified, underwent complete sequencing with the NGS technique. As shown in the Materials and methods section: "Only the individuals, diagnosed with breast and/or ovarian cancer without any of the 5 most common BRCA1 mutations, were qualified to the examination of BRCA1 and BRCA2 mutation carrier status, using the next generation sequencing (NGS)". In our opinion, it is clear that every qualified patient could get the screening BRCA test. If the test was negative and the patient met additional clinical criteria (also specified in the Materials and methods section: An affected individual was diagnosed with breast cancer or ovarian cancer and had at least 2 first- or second-degree relatives with the diagnosis of breast and/or ovarian cancer, and at least one of those cases had occurred before the age of 50. / An affected individual was diagnosed with breast cancer before the age of 50 or ovarian cancer at any age and had a first- or second-degree relative diagnosed with breast cancer (breast cancer in males) and/or ovarian cancer. / The same affected individual was diagnosed with both breast and ovarian cancer or bilateral breast cancer, including at least one case below the age of 50./ An affected individual was diagnosed with ovarian cancer and had at least one relative with breast cancer, diagnosed before the age of 50 or diagnosed with ovarian cancer.), they underwent NGS testing.
The Reviewer has suggested that NGS testing and complete sequencing of BRCA1 and BRCA2 were two different tests which was not the case, and it is stated in the Materials and methods section: The complete sequencing of the BRCA1 and BRCA2 genes, using the next-generation sequencing, was outsourced to subcontractors. However, some of the patients underwent
the third method: CNV detection by the MLPA method, which is also described in the Materials and methods section: [...]examining extensive deletions and duplications in the BRCA1 and BRCA2 genes, using the Multiplex Ligation Dependent Probe Amplification (MLPA)[...].
Review: Unfortunately, the author did not provide an explanation for this variation in testing methods.
The Materials and methods section of the manuscript precisely describes the qualification criteria for screening tests and NGS sequencing. In our opinion, the fact that not all the patients met all the NCCP criteria is clear enough to understand why some patients were not qualified for NGS testing but were qualified only for screening tests. The fact that specific qualification criteria were used is emphasized in the manuscript text: […] specific criteria for inclusion to genetic testing were applied.
Review: or clarify the differences in the results detected by these three approaches.
The exact differences are presented in the Tables in the Results section. The number of tests performed, the percentage, and the number of positive results are given in Table 1.
The mutations, detected by NGS that were not possible to detect with screening tests, are given in Table 2. In our opinion, the tables have appropriate and easy to understand headers.
The mutations, detected by the screening tests, were not additionally specified in the Results section, while they depended on the limitations of the PCR method, what is clearly shown in Tables 4 and 5 in the Materials and methods section. We believe that our way of data presentation is simple and comprehensive for readers with a basic knowledge of human genetics. We have added a paragraph in discussion that points out the differences in results in the scope of differences in methodology: the PCR-based screening tests focused on specific DNA regions. At the same time, the NGS enabled a simultaneous sequencing of longer DNA fragments, making possible multiple gene sequencing at one time, thereby substantially increasing the throughput and scalability figures.
Review: Additionally, the author failed to provide sufficient background information and did not adequately reflect on the significance, value, and guiding implications of the research findings.
Author comment: Following the issue, pointed by Reviewer, we have added a broader context to the Introduction section, which underlines the significance and implications of genetic testing for hereditary cancer. We have tried to keep it brief and comprehensible to scientists and medical professional working outside the topic of the paper: Cancer is a diverse group of diseases, being one of the most significant public health concerns worldwide. While most cancer cases are sporadic, a hereditary basis, linked to genetic mutations, occurs in a significant number of patients. The most common cases of hereditary cancer include the breast, ovary, colon, and uterus cancers. The understanding of the genetic basis of these cancers is crucial for risk assessment, early detection, and targeted treatment for the individuals more prone to these diseases. Genetic testing and counseling play a significant role in identifying individuals at increased risk and implementing appropriate surveillance programs. We believe that we reflect significantly on the implications of research findings in the Discussion section, while mentioning the key outputs:
- The introduction of the Program in 2018 was a breakthrough for genetic diagnostics in this area in Poland, and it also increased our knowledge about the role of genetic factors in the development of genetic diseases.
- The educational aspect of the Program is undoubtedly its key advantage, increasing the awareness of genetic testing and preventive measures among patients, what is an invaluable advantage.
- The results, presented in this study, indicate a fairly high percentage of detected pathogenic mutation carriers.
- It should, however, be noted that the diagnostic apparatus of the Program did not identify all the families at an increased cancer risk, which should be fairly possible, taking into account the today's state-of-the-art technological potentials.
- Considering the fact that the expansion of the diagnostic scope, specified in the Program, brought diagnostic results a minority of cases, it may be assumed that the real number of undetected carriers of pathogenic mutations may be significantly higher.
- Hopefully, future cost evaluations will take into account not only single cost areas but will also consider long-term clinical benefits, such as a higher percentage of effectively cured patients, more effective systemic treatment, shorter sick leave periods, and reduced cancer mortality rates.
- Additionally, the identification of such variants emphasizes the significance of functional studies which are rather rarely performed in cases of hereditary syndromes but are crucial to clarify the meaning of the variants detected.
Additionally we have added a paragraph in the Discussion section that emphasizes our recommendation to increase the presence of high-throutoput diagnostic tests in routine workflow, as well as point out the necessity of multidisciplinary approach: The hitherto practice of adhering to the international recommendations of scientific societies should be expedited and further supplemented by a comprehensive cost analysis, encompassing all the elements of personalized prevention, diagnostics, potential therapy, and rehabilitation, leading to health maintenance. A case in point could involve the augmentation (or omission) of specific molecular tests in favor of the high coverage next-generation sequencing procedure. Another pressing concern pertains to a horizontal integration, which more profoundly accounts for the collaboration among medical professionals of various specialties, involved in the prevention, diagnostics, and treatment of neoplastic ailments.
We also appreciate the grammar corrections, which were all implemented despite Line 45: "Out of whom" should be "Out of which." We believe that Out of whom was correct considering that we are refereeing to patients.

Reviewer 2 Report
Comments:
1. Since the title is "in Central Poland", please add info about central Poland such as map, and the detailed area.
2. The abstract does not have conclusion. Please add brief conclusion in abstract area.
3. Any data on p53 and MDM2 and FOXP3 on Tables 1and 2? Does it mean no p53 mutations found in this study?
4. For breast cancers, any data on triple negative vs ER+/PR+?
Author Response
Dear Reviewer,
We would like to thank you for your valuable comments. Below please find our comments to the issues raised in the review. All the changes have been made in manuscript in red. We are open to further improvement discussions, were the provided revisions not sufficient.
Review: Since the title is "in Central Poland", please add info about central Poland such as map, and the detailed area.
Author comment: We have added a section, dedicated to the detailed information about the residency of patients and localization of our medical facility: The patients, who attended our Clinic,were most often the residents of the Voivodeship of Lodz, where our facility is located. Due to the high diagnostic potential of our Clinic, we also received patients from the neighboring voivodeships, especially from Mazovia, Greater Poland, Silesia, Lower-Silesia, and Lesser Poland. We believe this section sufficiently describes the geography of the central Poland.
Review: The abstract does not have conclusion. Please add brief conclusion in abstract area.
Author comment: We have added section dedicated to the main results and key conclusion to improve the content of abstract: During a 46-month period, the objectives of the National Cancer Control Program (NCCP, pol. Narodowy Program Zwalczania Chorób Nowotworowych), coordinated by the Ministry of Health, were pursued by conducting genetic diagnostics in individuals at high risk of developing cancer. A total of 1024 individuals were enrolled
in the study, identifying 129 cases of germline mutations. The implementation of the NCCP identified genetic mutations in 4.43% of the patients qualified for BRCA1 and BRCA2 screening tests, 18.18% qualified for comprehensive NGS panel in cases of breast and ovarian cancer, and in 17.36% in cases of colorectal and endometrial cancer. The research conducted enabled to establish individualized preventive and therapeutic approaches to mutation carriers. However, the results prove that liberalizing the inclusion criteria for high-throughput diagnostics could significantly increase the percentage of detected carriers. This publication serves as a summary and discussion of the results obtained from the implementation of the NCCP, as well as of the role of genetic consulting in personalized medicine.
Reviews: Any data on p53 and MDM2 and FOXP3 on Tables 1and 2? Does it mean no p53 mutations found in this study?
Author comment: Yes indeed, there is no data about the alternation of p53 in Tables 1 and 2 because no pathogenic mutations in TP53 gene were detected. We have added a complete list of the tested genes in the Materials and methods section. The NGS panel did not include either MDM2 or FOXP3, while they are the modifiers of disease course and we focused on the genes correlated with hereditary cancer syndromes.
Reviews: For breast cancers, any data on triple negative vs. ER+/PR+?
Author comment: Unfortunately, we missed the histopathology data on a significant number of patients therefore we gave up showing the distribution of mutations, depending on the histological subtype. While this comment is highly important, we will undertake more effort to collect histopathology data in future projects because the mutational distribution, especially in cases of moderate risk genes and luminal subtypes, seems to be an interesting investigation topic.

Round 2
Reviewer 2 Report
No more comments
Author Response
Dear Reviewer,
We would like to thank you for your valuable comments. Below, please find our answers to issues raised in the review. All the changes we made in the manuscript after your comments are marked red. We are open to further discussion if the revisions are insufficient.
1. First of all, the most important thing is that I did not find any approval from an ethics committee, and I assume this is because these data come from participation in the Polish national screening program. It is very important that the authors specify if the people they analyzed were all enrolled in the Ministery screening program. If it is so, this must be clearly specified in the Materials and Methods section.
Then the authors claim: "In addition, we have presented the results of tests performed optionally, outside the program guidelines, and their effect on the percentage of detected mutation carriers." In my opinion, ethical approval is required for these data.
The authors state, "Informed consent was obtained from all subjects involved in the study." What informed consent are they referring to?
All the genetic testing was performed as a clinical service under standard clinical consent required for all germline genetic testing. In the case of patients tested in NCCP, the consent formula was provided by the Ministry of Health; in other cases, the interior declaration was used. Every clinical consent for genetic testing contains optional patient approval for anonymous publication of the results. The patients that did not give that approval were excluded from the study. Retrospective data collection from clinical testing does not require approval from the ethical committee, so we did not include it in Materials and Methods. However, while the report was prepared under the internal scientific grant, we do have approval from the Bioethics Committee of the Polish Mother’s Memorial Hospital Research Institute (approval number 80/2017). All the procedures performed in this study followed the principles of the Declaration of Helsinki. To clarify, we included the above information in the Material and Methods section.
2. The authors have omitted the correlation with patient molecular stratification data required by reviewer 2. They must explain why they have missed these data. In my opinion, these data are important and the authors could perform a correlation analysis with the data they have and add this information at least in the discussion section.
The correlation between the results of genetic testing with the breast cancer subtype suggested by reviewer 2 (Luminal A, Luminal B, TNBC) is not changing the course of genetic testing and consulting. Stratification of patients was done only by clinical criteria described in Materials and Methods. These are two reasons we did not collect data about the cancer subtype. Additionally, it would be a challenging logistic procedure while the patients were referred from different places with different histopathological departments. We suppose reviewer 2 was curious if our findings are consistent with the well-known fact that TNBC is more common in patients harboring germline pathogenic variants, especially in BRCA1 and BRCA2 genes. In this study, we focused on the genetic aspects of the treatment of cancer patients. It will not be possible to acquire histopathological data for all the patients included in our study, and we believe that doing partial analysis will not benefit the study and medical knowledge overall. If you believe it is necessary to present this data, independent of the above considerations, we need a much more extended period to carry out such analyses.
3. Finally, I believe that the authors should better stress the limitations and strengths of the present study in the conclusion section.
We have expanded the Conclusions to indicate the limitations and strengths of the study accurately.
Your sincerely,
Tadeusz Kałużewski